# Different Conditions during Confinement in Pasture-Based Systems and Feeding Systems Affect the Fatty Acid Profile in the Milk and Cheese of Holstein Dairy Cows

**DOI:** 10.3390/ani13081426

**Published:** 2023-04-21

**Authors:** Lucía Grille, Daniela Escobar, Maria Noel Méndez, María de Lourdes Adrien, Laura Olazabal, Víctor Rodríguez, Ronny Pelaggio, Pablo Chilibroste, Ana Meikle, Juan Pablo Damián

**Affiliations:** 1Departamento de Ciencias Veterinarias y Agrarias, Facultad de Veterinaria, Cenur Litoral Norte Universidad de la República, Paysandú 60000, Uruguay; 2Latitud-Fundación LATU, Montevideo 11500, Uruguay; descobar@latitud.org.uy (D.E.);; 3Departamento de Producción Animal y Pasturas, Facultad de Agronomía, Universidad de la República, Paysandú 60000, Uruguay; noemp21@gmail.com (M.N.M.);; 4Departamento de Desarrollo de Métodos Analíticos, Laboratorio Tecnológico del Uruguay (LATU), Montevideo 11500, Uruguay; 5Laboratorio de Endocrinología y Metabolismo Animal, Facultad de Veterinaria, Universidad de la República, Montevideo 13000, Uruguay; 6Departamento de Biociencias Veterinarias, Facultad de Veterinaria, Universidad de la República, Montevideo 13000, Uruguay; jpablodamian@gmail.com

**Keywords:** dairy products, lipid, bovine, grazing, total mixed ration, climatic condition

## Abstract

**Simple Summary:**

Pasture-based systems have advantages compared to confined systems, such as a higher proportion of beneficial milk fatty acids for their consumers. However, in mixed systems (grazing + total mixed ration), cows are more exposed to external climatic conditions. Due to climate change (high temperature or heavy rains), cows spend more time in confinement facilities where supplementation is offered, mainly in intensive mixed systems (high stocking rate). Therefore, the conditions of the facility during their confinement acquire great importance. To our knowledge, no studies have evaluated how the different facility conditions during confinement in a mixed system affect the fatty acid profiles in milk and cheese. The objective of this study was to compare the fatty acid profiles between mixed systems with different conditions during confinement (compost-bedded pack barns vs. outdoor soil-bedded pen) and mixed systems and confinement systems (100% total mixed ration) in a compost-bedded pack barns. In conclusion, the compost-bedded pack barns during confinement in a mixed system ensued a better milk quality (a higher percentage of Omega 3 (n-3) and C18:3 in the milk) compared to the outdoor soil-bedded pen. However, the fatty acid profiles in the milk, pooled milk (MilkP), and cheese were affected to a greater extent by the feeding management than by controlling the environment during the confinement.

**Abstract:**

The diet of dairy cows influences the fatty acid (FA) profiles of their milk and cheese, but how these are affected by different conditions during confinement in a mixed system (MS:grazing + total mixed ration:TMR) is not known. The aim of this study was to compare the FAs of the milk and cheese from MS in a compost-bedded pack barns (CB-GRZ) versus an outdoor soil-bedded pen (OD-GRZ) during confinement, and with a confinement system (100%TMR) in a compost-bedded pack barns (CB-TMR). Individual milk samples (*n* = 12 cows/group), cheese, and pooled milk (MilkP) samples were collected. The saturated FA percentages in the milk and the omega 6/omega 3 ratio in the MilkP and cheese were greater for the CB-TMR (*p* < 0.0001), while the unsaturated and monounsaturated FA percentages in the milk were lower for the CB-TMR than the MS (*p* < 0.001). The milk n-3, C18:3, and conjugated linoleic acid percentages were lower for the CB-TMR than the MS (*p* < 0.001). The milk n-3 and C18:3 were higher for the CB-GRZ than the OD-GRZ (*p* < 0.01), but no differences were observed between the MS in the MilkP and cheese. In conclusion, CB-GRZ cows during confinement produced better quality milk compared to OD-GRZ cows. However, the FA profiles of the milk, MilkP, and cheese were affected to a greater extent by the feeding management than by the conditions during confinement.

## 1. Introduction

Milk fat is one of the main components that determines the nutritional value of milk for humans, due to its energetic content and the bioactive properties of some of its fatty acids for human health [1,2]. The fatty acid profile (FAP) of milk is characterized by a higher proportion of saturated fatty acids (SFA) and a lower proportion of monounsaturated fatty acids (MUFA) and polyunsaturated fatty acids (PUFA) [3]. The fat content and FAP are the most variable components of milk. One of the factors that most influences its variability is the nutritional management of the cow [4,5,6,7]. Pasture-based feeding systems or the inclusion of fresh pasture in the diet of confined dairy cows improves the FAP of milk, benefiting human health [8,9]. In this sense, in pasture-based systems, the milk’s SFA content (C12:0, C14:0, and C16:0) and omega 6/omega 3 (n-6/n-3) are lower and some PUFAs are higher than those in TMR systems, such as 18:3 (n-3) and C18:2 (cis9-trans 11; conjugated linoleic acid: CLA), which is essential for humans [7,10,11,12,13,14]. The differences generated by changes in a cow’s diet that have reported in milk FAPs are also reflected in cheeses. Higher concentrations of vaccenic acid (VA), CLA, and C18:3 (n-3) have been obtained from cheeses elaborated from pasture-based milk systems [15,16,17,18]. Therefore, pasture inclusion in a dairy cow’s diet improves the nutritional profile of the FA in its milk and cheese, due to an increase in some of the FAs that are beneficial for human health, such as CLA, VA, and C18:3 (n-3), and a reduction in the n-6/n-3 ratio.

Cows are exposed to environmental and seasonal weather changes within pasture-based systems [19]. In countries with temperate climates, cows graze throughout the year, resulting in a better nutritional quality of their milk than that in confined systems (i.e., [9,20]). However, climate change has caused more extreme weather events, such as increased temperatures in summer, droughts, frosts in spring, and heavy rains, which affect the growth of the pastures and the time available for grazing. Therefore, these climatic phenomena become more important in temperate countries [21,22,23]. Extreme weather conditions can affect the physiology (an energy demand for thermoregulation, a decrease in dry matter intake, and a decrease in milk production) and behavior of dairy cows [24,25,26,27]. Cows can suffer from heat stress during spring–summer, which impacts their milk production and its quality. It has been reported that heat stress (a temperature humidity index (THI) above 68) negatively impacts their welfare [28], milk production, and its yield and fat content [29,30,31,32], as well as the milk’s FAP [33]. Salamon et al. [34] showed higher percentages of C18:2, C18:3 (n-3), C18:1 (Oleic), and CLA in milk obtained during summer compared to those in milk from winter. In addition, lipid catabolism in thermally stressed dairy cows leads to an increased plasmatic FA content (as linolenic and oleic), and consequently, increased long-chain FAs and decreased short- and medium-chain fatty acids in the milk [33]. Furthermore, higher levels of MUFAs, PUFAs, and CLAs and lower n-6/n-3 ratios and palmitic acid concentrations (C16:0) in milk and cheese have been reported during spring compared to winter and autumn [35,36,37]. A possible explanation for this may be the proportion of conserved forages in the diet (mainly autumn), which have a poor fat quality compared to spring diets; thus, more fresh pasture is generally included in these diets [22,36]. In addition, in pasture-based systems, the season determines the rate of the grass growth, which affects the quality, digestibility, and consumption of the grass [38]. Therefore, in pasture-based systems, the season of the year could impact the FAP and cheese quality.

Uruguayan dairy systems are characterized by pasture-based systems (pasture 55%, supplementation with conserved forage 19%, and concentrates 25%) [22]. Unlike confined systems, pasture-based systems have the advantage of cows being able to display behaviors that are typical of their species, which therefore improves the animal welfare, but they are also more exposed to extreme environmental conditions, such as heavy rains and mud in winter or heat stress in summer, which could also affect their welfare [39,40]. Most dairy systems in Uruguay keep cows confined in open sky paddocks when they are not grazing, without shade over the resting and feeding areas [41]. Pasture-based system cows are supplemented in open areas without environmental protection (i.e., shade and a roof), and only 22% of dairy farms provide supplementation in specific feeding facilities [41]. However, over the last few years, confinement (places where cows are kept while they are not in grazing) has changed. There are many farms where confinements have fully covered sheds, drinkers, feeders, and different types of bedding. These confinements are characterized because the cows have little exposure to the external climate (a better control of the environment and better comfort). Some examples of these are the systems called “compost barns” or “compost-bedded pack barns” [42,43].

The milk and cheese quality from different feeding systems (confinement vs. pasture) and different seasons have been studied, but confounding effects between the environmental factors and feeding systems are evident [16,44,45]. To our knowledge, there are no studies that have evaluated how different facility conditions (compost-bedded pack barns vs. outdoor soil-bedded pen) during confinement in a mixed system (MS; under the same feeding condition: grazing + TMR) influence the milk fat quality and affect the dairy products’ quality (i.e., cheeses). Therefore, firstly, we hypothesized that both MSs would achieve a healthier FAP in the milk and cheese from spring calving cows compared to the confined system (CB-TMR), although we predicted a lower milk and fat yield. Secondly, we hypothesized that different facility conditions during confinement in an MS (compost-bedded pack barns vs. outdoor soil-bedded pen) during summer could impact the FAP in the milk and cheese from spring calving cows. The aim of this study was to compare the milk and cheese FAPs from MSs with different facility conditions during confinement (compost-bedded pack barns vs. outdoor soil-bedded pen), and compare them with a confinement system (100% TMR) in spring calving cows from a compost-bedded pack barns.

## 2. Materials and Methods

### 2.1. Location, Animals, and Treatments

The experimental protocol was evaluated and approved by the Comisión Honoraria de Experimentación Animal (CHEA), Universidad de la República, Montevideo, Uruguay (ID 682; Exp: 020300-000602-18). The study was conducted at the Estación Experimental ‘Dr. M. A. Cassinoni’’-Facultad de Agronomía Paysandú, Uruguay (Location 32°22′49′′ S, 58°03′04′′ W).

In total, thirty-six spring calving Holstein cows with 2.7 ± 1.2 lactation and 620 ± 64 kg of body weight were used. The average calving date was 16 August 2019 ± 8.2 d. All the cows were under the same management and feeding conditions throughout the 21 d before the expected calving date. The cows were blocked by their calving date, number of lactations, pre-calving body conditions, and body weight, randomly assigned to one of the following treatments immediately after calving, and grouped to 4 cows/pen. This constituted three treatments: (1) cows that were fed with TMR ad libitum in compost-bedded pack barns (CB) facilities (CB-TMR, *n* = 12) (a low environment exposure); (2) mixed-system (MS) cows in CB facilities (a low environment exposure, same barn as previous) during confinement (CB-GRZ, *n* = 12); and (3) mixed-system cows in outdoor soil-bedded pens during confinement (a high environment exposure) (OD-GRZ, *n* = 12).

In the CB-TMR and CB-GRZ systems, the cows were confined within a fully roofed barn with CBs, fans, and sprinklers (activated when the temperature exceeded 25 °C) and protected from solar radiation, rain, mud, and wind (a low environment exposure). Each pen was 6 m wide by 13.5 m long (9 m of bed and 4.5 m of feeder space). In the OD-GRZ treatment, the cows were confined in an outdoor soil-bedded pen with shade. The outdoor soil-bedded pens (OD) consisted of 2 paddocks (which alternated according to the mud conditions), each with the following measurements: 5.2 m wide by 46 m long (shaded area: 4.8 m^2^/cow). The water was disposed of ad libitum for all the treatments.

The total mixed rations were composed of whole plant sorghum or corn silage (CP: 5.9, 5.8%; NDF: 51.5, 41.7%; and ADF: 30.7, 22.5%; respectively), rye grass silage (CP: 10.9%; NDF: 49.6%; and ADF: 29.9%), fescue hay (CP: 9.7%; NDF: 68.3%; and ADF: 37.4%), and a commercial concentrate mix (CP: 25.5%; NDF: 34.5%; and ADF: 13.5%). Their chemical composition is presented in Table 1. The TMRs were formulated according to the National Research Council [46], for a body weight of 650 kg and 45 L/d of milk production (4% milk fat) for the CB-TMR treatment. In the MSs’ (CB-GRZ and OD-GRZ), the supplementation with TMRs was adjusted to ensure that the cow requirements and productivity goals were met. In the CB-TMR treatment, the cows were fed ad libitum twice a day (5:30 am and 8:00 pm). In the mixed systems, TMRs were offered once a day at 8:00 am from November to April. 

The pasture was composed of tall fescue (*Festuca arundinacea*) (October–spring), soy (*Glycine max*) (January–summer), and lucerne (*Medicago sativa*) + orchard grass (*Dactylis glomerata*) (March–late summer). The herbage allowances (HA) were 28 kg DM/cow/day, 46 kg DM/cow/day, and 31 kg DM/cow/day, respectively. The herbage mass (kg DM/ha) at the ground level was estimated weekly using the double sampling technique adapted from Haydock and Shaw [47]. The method was calibrated fortnightly using a 5-point scale with 3 replicates for each point. The herbage allowance was then determined, adjusting the daily strip area for grazing. The grazing occurred in weekly occupation plots. 

The THI values and precipitation from 10 d prior to the milk and cheese sampling are shown in Figure 1. The temperature humidity index was calculated according to Bernabucci et al. [29].

### 2.2. Experimental Design

Based on the times of the sample collection (spring and summer vs. late summer) and the type of the samples (individual milk vs. pooled milk and cheese), the study was divided into two sections. The first section was based on individual cow milk during spring and summer, and the second section was based on polled milk and cheese during late summer (Figure 2).

#### 2.2.1. Individual Cow Milk during Spring and Summer

All the cows were milked twice a day (4:00 am and 5:00 pm) and the individual milk yields were individually recorded. 

The fat-corrected milk (FCM) yield was standardized at 3.5% fat and calculated according to Pastorini et al. [48] and Mendoza et al. [49]

The total mixed rations, pasture samples (Table 1), and individual milk samples (the composite sample representative of both daily milkings) were collected at two moments of lactation: ML1: 80 ± 15 days in milk (DIM; October: spring; early lactation) and ML2: 155 ± 15 DIM (January: summer; mid-lactation) 

#### 2.2.2. Pooled Milk and Cheese Manufacture during Late Summer

For the cheese making, pooled milk (MilkP) was collected from both milkings (am and pm) in each pen into 60 L buckets. The trials were performed over two consecutive weeks during March (late summer: 210 ± 15 DIM). The pooled milk was obtained from 2 pens for each treatment in each week. This resulted in a total of 12 cheeses being manufactured in the experiment (each treatment: *n* = 4). This study followed a similar methodology to other studies, with the production system having effects on the characteristics of the cheese [16,44,45]. Unlike the aforementioned studies, ours used more experimental units. The pooled milk was stored and transported under refrigerated conditions (4 °C) to the Latitud–Fundación LATU pilot plant (Montevideo, Uruguay). The MilkP was standardized to 3.0% fat in a Westfalia Separator skimmer model LWA 205-1 (Oelde, Germany). Dambo-type cheeses were produced, as previously described by Escobar et al. [50]. Briefly, the cheese making process was carried out in a 50 L double O vat with a double jacket and a controlled mechanical agitation and cutting system. The milk was pasteurized at 72 °C for 15 s when the milk had cooled to 32 °C and CaCl^2^ (Promilk^®^, Arras, France) and a mesophilic homofermentative starter culture (CHR HANSEN R 704, Christian Hansen, Denmark) were added. Thirty minutes later, the coagulant 100% chymosin (Maxiren^®^, Hørsholm, Denmark) was added, and the cultured milk was allowed to set for approximately 30 min. The curd was cut until a size of “corn grain” was obtained, which was cooked until 42 °C (stirring continuously). After this cooking, the whey was drained off, and the curd was distributed into a 1 kg polypropylene cylinder mold and pressed for 3 h (with intermediate rotations). The acidification of the cheese was measured after the pressing, and the pH was checked until it reached the final pH of 5.40 ± 0.1, where it was kept at 4 °C for 16 h. Afterward, the cheeses were salted in brine (19 °B) for 5 h. Then, they were kept in a chamber at 4 °C ± 2 for 24 h, after which, the cheeses were vacuum packed. The cheeses were kept in a ripening chamber at 12 °C for 30 d until the analysis.

The total mixed rations and pasture were taken on the same day of the milk collection (MilkP) for the cheese manufacture (Table 1). The MilkP samples for the FAP analysis were taken from the pilot plant (LATU) after the pasteurization (before cheese manufacture) and the cheese samples were taken after 30 d of ripening.

### 2.3. Samples Analysis

#### 2.3.1. Fat Yield

The milk fat content (%) was determined using LactoScope FT infrared (FTIR) (Delta Instruments, Drachten, The Netherlands) at the COLAVECO in Colonia Suiza, Uruguay.

#### 2.3.2. Fatty Acid Profile in Milk (Individual and Pooled) and Cheeses

The milk fat was extracted according to Hara and Radin [51] and the cheese fat was extracted according to Folch et al. [52]. FA methyl esters were prepared via the trans-methylation procedure described by Mossoba et al. [53]. These fatty acid methyl esters were quantified using a gas chromatograph (Agilent Technologies 6890, Palo Alto, CA, USA) and a mass spectrometer (Agilent Technologies 5973), as described by Grille et al. [7]. The samples were run in duplicate and the FAME standards (Supelco 47885-U, Bellefonte; 37 FAME from C4:0 to C24:0) were analyzed at regular intervals for quality control purposes and to determine the recovery and correction factors for the individual FAs. The intra- and inter- assay coefficients of variation for each measured analyte were, on average, 3% and 6%, respectively. The milk fat compositions were expressed in grams of each individual FA per 100 g of the total FAs. The de novo FAs were considered to be the sum of the C4:0 to C15:1 FAs, the mixed FAs to be the sum of C16 and C16:1, and the preformed FAs to be a sum of >C17:0. The Atherogenicity and Thrombogenicity indexes (AI and TI) in the milk were calculated as described by Ulbricht and Southgate [54]. The Atherogenicity index (AI) was calculated as (12:0 + 4 × 14:0 + 16:0/(MUFA + PUFA) and the Thrombogenicity index (TI) was (C14:0 + C16:0 + C18:0) (0.5 × MUFA + 0.5 × (n − 6) +3 ×(n − 3)/(n − 6). The FA analysis was performed at the Technological Laboratory of Uruguay (LATU) in Montevideo, Uruguay.

#### 2.3.3. Pasture and TMR Chemical Composition and Fatty Acid Profile

The pasture and TMRs were weighed and oven dried at 55 °C for 48 h, in order to be later analyzed to determine their dry matter (DM), crude protein (CP), neutral detergent fiber (NDF), and acid detergent fiber (ADF) contents, according to AOAC (2000) [55]. The total nitrogen for the CP estimation was analyzed with the Kjeldahl method according to AOAC (1984), as were the digestion with sulfuric acid and the subsequent distillation and titration [56]. The neutral detergent fiber used α-amylase and both the NDF and ADF were measured using an ANKOM200 Fiber Analyzer (ANKOM Technology Corp., Fairport, NY, USA), according to Méndez et al. [23]. The lipids were extracted according to Folch et al. [52], and the FAP was performed as previously described for the milk and cheese samples.

### 2.4. Statistical Analysis

The milk yield (kg/d), fat yield (kg/d), fat content (%), and FAP (%) from Section 1 were analyzed with an ANOVA for repeated measures using the GLIMMIX of SAS (SAS Studio, Shinjuku, Tokyo). The Shapiro test was used to evaluate the normal distribution of the data. The goodness of fit for each model was checked by a visual inspection of the residuals. The election of the final models considered the Akaike’s information criterion (AIC). The statistical model for the milk yield, fat (content and yield), and FAP included the treatments (CB-TMR, CB-GRZ, and OD-GRZ), moments of lactation (ML1 and ML2), and interactions between the treatments and moments of lactation as fixed effects, and the cow was considered as a random effect within each treatment. For the individual milk data (Section 1), the experimental unit was the cow, while for the MilkP and cheese (Section 2), the pen was considered as the experimental unit and the treatment was considered as a fixed effect. In Section 2, the week was considered as a random effect. In all the variables, the post-calving days were included as co-variables. Post hoc comparisons were performed with the Tukey–Kramer test. For a better visualization of the results, when the data sets from both moments of lactation (ML1 and ML2) in each treatment were analyzed, they were expressed as the average of the treatments (AT). In total, three treatments (CB-TMR, CB-GRZ, and OD-GRZ) were analyzed and the data sets from each moment of lactation were expressed as the average lactation moment (AML), either for the M1 (AML1) or M2 (AML2). The results were considered to be significant with an alpha of ≤0.05 and trends between 0.05 and 0.10. The data are presented as mean ± SEM (standard error of the mean).

## 3. Results

### 3.1. Milk Yield and Fat (% and kg/d)

There were treatment and moment of lactation effects on the milk and fat yields (kg/d), but the interactions between the factors were not significant (Table 2). The milk and fat yields were greater for the CB-TMR treatment than for the CB-GRZ and OD-GRZ treatments (*p* < 0.001), and there was no difference between both the MSs (CB-GRZ and OD-GRZ). The milk yield was higher for AM1 than AM2 (*p* = 0.03) and the fat yield tended to be higher for AML1 than AML2 (*p* = 0.07). There were no treatment and moment of lactation effects and there was no interaction between the treatments and moments of lactation in the milk fat content (%) (Table 2).

#### 3.1.1. Individual Cow Milk during Spring and Summer

There was a treatment effect (*p* < 0.05) on the SFAs, UFAs, and MUFAs, n-3, n-6, and n-6/n-3, trans, mixed, and preformed FA content, and atherogenicity (AI) and thrombogenicity indexes (TI) of the milk (Table 3). The saturated FAs, n-6, n-6/n-3 ratio, AI, and TI were greater for the CB-TMR treatment than the CB-GRZ and OD-GRZ treatments (*p* < 0.001), and CB-TMR had lower UFAs, MUFAs, and trans FAs than CB-GRZ and OD-GRZ (*p* < 0.001). Additionally, CB-TMR had a lower n-3 than the milk from the CB-GRZ cows (*p* = 0.0001), but no difference was observed for OD-GRZ. Between the mixed systems, CB-GRZ had a greater n-3 than OD-GRZ (*p* = 0.005). There was moment of lactation effect on the SFAs, UFAs, MUFAs, n-6, novo FAs, mixed FAs, AI, and TI (Table 3). The saturated FAs, n-6, de novo, AI, and TI decreased from AML1 to AML2 (*p* < 0.01). The MUFAs, UFAs, and mixed FAs increased from AML1 to AML2 (*p* < 0.001). The polyunsaturated FAs tended to be greater at AML1 than AML2 (*p* = 0.08). There was an interaction of the n-3, n-6, and mixed FAs between the treatment and the moment of lactation (Table 3). In ML1, CB-GRZ had a greater n-3 than CB-TMR and OD-GRZ (*p* = 0.02; *p* = 0.004). For ML2, CB-GRZ had a greater n-3 than CB-TMR (*p* = 0.04), while OD-GRZ was intermediate. None of the three treatments showed differences in their n-3 values between ML1 and ML2 (Table 3). The milk of CB-TMR had a greater n-6 than that of CB-GRZ and OD-GRZ (*p* < 0.001) at ML1 and ML2, and no differences were observed between the mixed systems at both moments of lactation. The mixed FAs at ML1 were higher for CB-TMR than CB-GRZ and OD-GRZ (*p* < 0.0001), which were similar to each other. At ML2, no differences were found between the treatments for the mixed FAs (Table 3). The main changes in the SFAs were due to C4:0; C6:0; C8:0; C10:0, and C12:0, which were greater for CB-TMR than CB-GRZ and OD-GRZ, but no differences were observed between both MSs (Table 4). Regarding C14:0 and C16:0, there was an interaction between the treatment and moment of lactation (Table 4). At ML1, the C16:0 in the milk was greater for CB-TMR than the MS (*p* < 0.001), but the C14:0 was lower for CB-TMR and CB-GRZ than OD-GRZ (*p* = 0.04). At ML2, no differences were found between the treatments for any FAs. Regarding PUFAs, C18:2, CLA, and C18:3 had a treatment effect (Table 3 and Table 4). Both FAs were greater for CB-GRZ and OD-GRZ than CB-TMR (*p* < 0.0001). Furthermore, the C18:3 was greater for CB-GRZ than OD-GRZ (*p* = 0.001) (Table 4).

#### 3.1.2. Fatty Acid Profile in Pooled Milk and Cheese

A treatment effect was observed for the n-3, n-6, and n-6/n-3 ratios in the MilkP and cheese (Table 5). The MilkP and cheese from the CB-TMR treatment had a greater n-6 and n-6/n-3 ratio than the CB-GRZ and OD-GRZ treatments (*p* < 0.01). Regarding n-3, for the MilkP, CB-TMR was lower than that of CB-GRZ and OD-GRZ (*p* < 0.0001). In the cheese, the n-3 was lower for CB-TMR than CB-GRZ (*p* = 0.03), but no difference was found between CB-TMR and OD-GRZ. There was no difference in any of the FAs between the mixed systems for both the MilkP and cheese (Table 5).

## 4. Discussion

In this study, it was evidenced that the main changes in the individual milk, MilkP, and cheese were observed between the confined system (CB-TMR) and mixed systems (CB-GRZ, and OD-GRZ). However, the differences in the facilities during the confinements in the mixed systems (compost-bedded pack barns vs. outdoor soil-bedded pen) affected some of the FAs in the milk which are beneficial to humans. 

The CB-TMR system had a higher milk yield and fat yield (kg/d) compared to both the mixed systems, which is in accordance with the findings of Salado et al. [57] and Bargo et al. [58]. This difference in the fat yield (kg/d) was due to the higher milk yield in CB-TMR, since there were no differences in the fat contents (%) between the treatments. The milk from CB-TMR (Section 1), as well as the MilkP and cheese from this treatment (Section 2), showed clear differences in their fatty acid profiles compared to those from the mixed systems (CB-GRZ and OD-GRZ), as evidenced by a higher percentage of UFAs, MUFAs, C18:2 (CLA), and C18:3 (n-3), as well as a lower percentage of SFAs and a lower n-6/n-3 ratio. Given that low concentrations of UFAs, MUFAs, C18:2 (CLA), and C18:3 (n-3), a higher percentage of SFAs, and a higher n-6/n-3 ratio are associated with less healthy milk [59,60], our results show that the fatty acid profiles in the milk from CB-TMR are less healthy than those from the mixed systems. In addition, the mixed-system milk (at both moments of lactation) had better atherogenicity and thrombogenicity indexes compared to that from CB-TMR. For the MilkP and cheese, those from the CB-TMR treatment had a lower n-3, a higher n-6, and a higher n-6/n-3 ratio than those from the MSs. The fact that the MS cows had a better fatty acid profile in their milk, MilkP, and cheese compared to the CB-TMR cows (in favor of the beneficial FAs for human health) could be due to the greater contribution of the precursors to the milk FAs, such as C18:2 and C18:3, which are provided by the pasture [2,8]. These results confirm what has been reported in other studies, where the milk and dairy products from mixed systems (grazing + TMR) present higher proportions of MUFAs and PUFAs, lower atherogenic risks and SFAs, a lower n-6/n-3 ratio (the recommended value should be below 4/1; Simopoulos [61]), AI, and TI, and a higher content of healthy FAs (i.e., n-3), from the point of view of human health, compared to TMR-fed systems [7,9,20,35,48,62]. In addition, although the cows from the pasture-based systems had a better milk fat quality, this was to the detriment of the milk’s fat yield. In this sense, the lower milk and fat yields in the mixed-system cows could be due to the lower dry matter, lower energy intake, and higher energy requirements during grazing, in comparison to the cows fed with TMRs ad libitum in confinement, which resulted in a lower productive performance [58,63,64]. Therefore, our results highlight the importance of pasture consumption for cows in mixed systems, presenting a better-quality fatty acid profile in their milk and cheese compared to those without pasture consumption (100% confined systems), despite its detriment to the productive variables.

Regarding the effect of environmental exposure (low and high) during confinement on the milk fatty acid profiles, differences between the mixed systems were only found at ML1 (spring, Section 1), but no differences were found between the mixed systems for the MilkP and cheese in late summer (Section 2). In spring (ML1), when the OD-GRZ cows were exposed to rain during confinement (Figure 1), a negative impact on their milk fatty acid profiles was evidenced by the lower content of n-3 in their milk, in contrast to the CB-GRZ cows. It has been reported that, when the facility conditions are not comfortable enough, cows modify their behavior, with less time lying and more time standing during confinement [65,66,67,68], which could later alter their grazing behavior in order to recover the required resting time for their welfare [69,70]. In fact, this is consistent with the results obtained by our team (in the same experiment), in which it was observed that, when extreme external conditions occurred (accumulated rain and mud), the OD-GRZ cows were lying less frequently during confinement, but more frequently during grazing than the CB-GRZ cows [65]. Meanwhile, it is possible to speculate that these changes in their ingestive behavior during grazing could have altered their dry matter intake, finally impairing the milk fat quality (a decrease in n-3). Therefore, a lower exposure to the environment during confinement (CB-GRZ) in a mixed system (confined in a fully roofed barn with CB, protected from climatic conditions outside) improved some of the parameters in the milk (C18:3 and n-3), thus making it healthier for human consumption. Contrary to our original hypothesis, we did not find any differences in the milk FAPs between both the mixed systems during the summer (ML2). Several studies have reported that heat stress in summer affects milk production, yield, and fat content, as well as milk fatty acid profiles [29,30,31,33,71,72]. Therefore, it was expected that, during summer, the OD-GRZ cows, being more exposed to the environment during confinement, would present a greater heat stress, resulting in a negative impact on their milk quality and FAs. However, such differences in the milk fat quality between the mixed systems were not found, which could be associated with the particular weather during that summer and the good conditions of the outdoor soil-bedded pen used in our study. Therefore, this lack of difference could be due to: (i) no high THI (as would be expected in the region during this season) or rainfall values during the days prior to sampling; or (ii) the OD-GRZ facilities having enough shade (according to the number of animals per pen) and water ad libitum, which could have contributed to mitigating the negative effects of the summer on the OD-GRZ cows, therefore preventing a negative effect on the milk fat quality. Román et al. [40] observed that heat stress mitigation strategies, such as a pen with shade (even with a THI below 72), improved the performance and behavior of dairy cows. Based on all of the above, under the conditions carried out in this study and the environmental particularities of summer and the outdoor soil-bedded pens, the OD-GRZ cows probably did not present negative repercussions due to heat stress; thus, there were no changes in their milk.

Regarding the MilkP and cheese, we anticipated that, if the OD-GRZ cows suffered more heat stress during the summer, this higher stress response would have had a greater negative impact not only on the milk, but also on the final product, cheese. However, in line with what was mentioned in Section 1 (with the individual milk during summer), the greater exposure to the environment during confinement in a mixed system did not affect the fatty acid profiles of the milk and cheese. Probably, as we commented before, the good conditions of the outdoor soil-bedded pens and the characteristics of the environment in the moments close to the sampling did not generate great enough changes in the animals to impact the quality of their milk and cheese, even with a THI above 72. In addition, the fact that the cheeses were made during late lactation could have influenced the lack of differences found in the fatty acid profiles between the mixed systems, given that heat stress in late lactation would have had a lower effect on the solids than that in early lactation [40]. Furthermore, it is also important to consider that, in Section 2, the experimental unit was smaller than that in Section 1; therefore, we are cautious in saying that there were no differences. Future studies using a larger number of experimental units are needed to confirm whether exposure to the environment during confinement in a mixed system affects the FAs in pooled milk and cheese. In any case, in Section 2, it is highlighted that the changes in the milk fatty acid profiles between the treatments were reflected in the fatty acid profiles of the cheese; therefore, the cheeses reflected and retained the fat quality of the milk.

## 5. Conclusions

The most important changes in the FAPs of the milk, MilkP, and cheese were due to the presence of pasture in the cows’ diet, rather than a difference in the facilities during confinement in an MS (compost-bedded pack barns vs. outdoor soil-bedded pen), which was evidenced by the higher UFAs and MUFAs and lower SFAs and n-6/n-3 ratio in the milk, and the higher n-3 and lower n-6 and n-6/n-3 in the MilkP and cheese from the MSs than those from the CB-TMR treatment. Nevertheless, the compost-bedded pack barns conditions presented a better quality of milk fat (a higher percentage of n-3 and C18:3 in the milk) compared to that from the cows in the outdoor soil-bedded pen.

## Figures and Tables

**Figure 1 animals-13-01426-f001:**
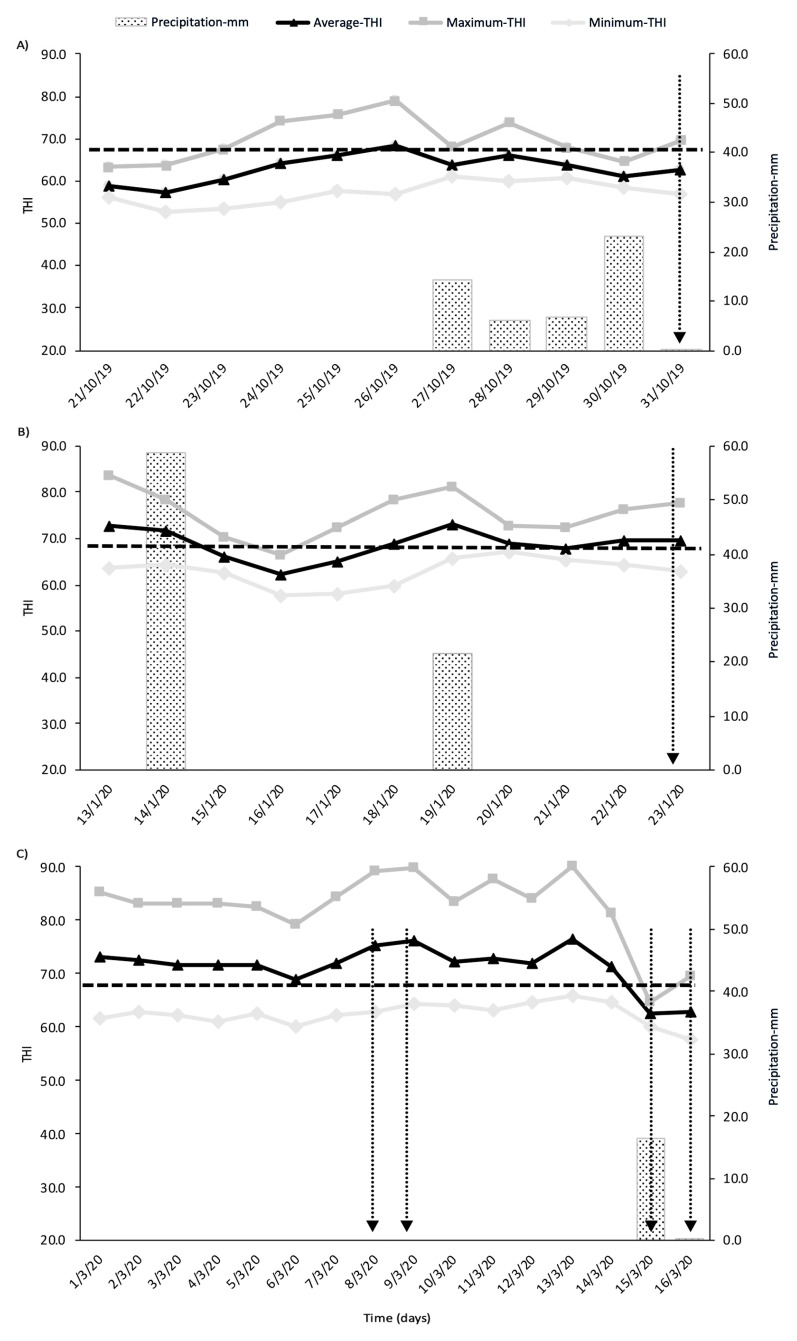
Temperature humidity index (THI) (average, maximum, and minimum) and precipitation (mm) (**A**) 10 days before milk sampling in spring (ML1); (**B**) 10 days before milk sampling in summer (ML2); and (**C**) late summer (MilkP sampling for cheese manufacture). Dotted arrow indicates sampling time.

**Figure 2 animals-13-01426-f002:**
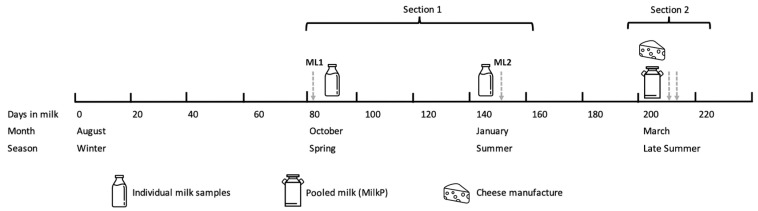
Experimental design. Section 1: Individual cow milk during spring and summer. Section 2: Pooled milk and cheese manufacture during late summer.

**Table 1 animals-13-01426-t001:** Chemical compositions and fatty acid profiles in pasture and total mixed rations (TMR) during Section 1: individual milk samples in both moments of lactation (ML1 and ML2) and Section 2: MilkP and cheese manufacture.

Individual Milk Samples					Cheese Manufacture	
Section 1					Section 2	
	ML1		ML2			
	TMR	Pasture	TMR	Pasture	TMR	Pasture
CP (% DM)	21.6	12.2	20.1	14.2	16.4	21.25
NDF (% DM)	34	59	33	26	26	44
ADF (% DM)	19	30	19	19	15	20
FAs (g/100 g)						
C8:0	0.07	nd	0.11	nd	0.5	0.7
C10:0	0.04	1.56	0.02	nd	1.1	1.5
C12:0	0.35	0.05	0.24	0.55	1.4	1.9
C14:0	0.74	0.57	0.59	0.65	4.3	5.6
C14:1	0.08	0.7	0.08	0.11	0.3	0.5
C15:0	0.07	3.17	0.08	0.83	0.5	0.7
C16:0	31.95	20.74	28.27	18.09	21.6	25.1
C16:1	0.47	2.19	0.49	1.93	0.8	0.9
C17:0	0.15	0.37	0.12	0.24	0.4	0.6
C17:1	0.03	0.03	0.04	nd	0	0.1
C18:0	3.51	2.04	2.99	5.23	4.8	6.9
C18:1cis	29.71	5.7	34	4.17	32	14.9
C18:1trans	0.19	nd	0.22	nd	0.9	1.7
C18:2cis	24.34	12.63	24.3	17.25	23.9	12.1
C18:3 (n-3)	3.86	32.51	3.97	39.1	4.3	18.5
C18:3 (n-6)	0.03	0.18	0.06	0.13	.	.
C20:0	0.45	0.77	0.44	1.05	0.3	0.4
C22:0	0.15	2.56	0.42	1.78	0.3	0.3
C23:0	nd	nd	nd	nd	0.4	1.5
C24:0	0.35	0.89	0.37	2	0.2	0.2
Ether extract (g/100)	3.5	1.6	4.1	1.5	3.5	2.4

ML1: 80 ± 15 days in milk (DIM), ML2: 155 ± 15 DIM. CP: crude protein; NDF: neutral detergent fiber; and ADF: acid detergent fiber. nd: not detected.

**Table 2 animals-13-01426-t002:** Milk yield and fat (mean ± SEM) in confined (CB-TMR) and mixed systems with low (CB-GRZ) and high (OD-GRZ) environmental exposure in spring calving cows at two moments of lactation (ML1 and ML2).

	CB-TMR			CB-GRZ			OD-GRZ					*p* Value		
	ML1	ML2	AT	ML1	ML2	AT	ML1	ML2	AT	AML1	AML2	T	ML	T * ML
Milk yield (kg/d)	41.3 ± 1.0	39.4 ± 1.0	40.7 ± 0.7 ^x^	29.9 ± 1.1	27.7 ± 1.1	28.3 ± 0.7 ^y^	28.7 ± 1.0	26.5 ± 1.0	27.5 ± 0.7 ^y^	32.9 ± 0.6 ^A^	31.1 ± 0.6 ^B^	<0.001	0.03	ns
3.5% FCM (kg/d)	39.5 ± 1.1	36.5 ± 1.1	38.5 ± 0.8 ^x^	26.8 ± 1.2	25.9 ± 1.2	26.4 ± 0.8 ^y^	27.4 ± 1.2	25.7 ± 1.2	26.6 ± 0.8 ^y^	31.4 ± 0.7 ^A^	29.5 ± 0.7 ^B^	<0.001	0.04	ns
Fat														
(%)	3.4 ± 0.1	3.2 ± 0.1	3.4 ± 0.1	3.1 ± 0.2	3.2 ± 0.2	3.1 ± 0.1	3.3 ± 0.2	3.3 ± 0.2	3.3 ± 0.1	3.3 ± 0.1	3.2 ± 0.1	ns	ns	ns
(kg/d)	1.4 ± 0.05	1.2 ± 0.05	1.3 ± 0.04 ^x^	0.9 ± 0.05	0.8 ± 0.05	0.9 ± 0.04 ^y^	0.9 ± 0.05	0.8 ± 0.05	0.9 ± 0.04 ^y^	1.05 ± 0.03	0.98 ± 0.03	<0.001	0.07	ns

T: treatment; ML: moments of lactation; and T * ML: interaction between treatments and moments of lactation. Treatments: CB-TMR, CB-GRZ, and OD-GRZ. ML1: 80 ± 15 days in milk (DIM), and ML2: 155 ± 15 days in milk (DIM). AT: average from both moments of lactation in each treatment; AML1: average from treatments (CB-TMR, CB-GRZ, and OD-GRZ) in ML1; and AML2: average from treatments (CB-TMR, CB-GRZ, and OD-GRZ in ML2). Fat-corrected milk standardized at 3.5% fat (3.5% FCM). Different letters indicate significant difference (*p* < 0.05): ^x,y^ indicate significant difference between AT; ^A,B^ indicate significant difference between AML. ns: no significance. 3.2. Milk fatty acid profile.

**Table 3 animals-13-01426-t003:** Fatty acid profile (FAP) (mean ± SEM) in confined (CB-TMR) and mixed systems with low (CB-GRZ) and high (OD-GRZ) environmental exposure in spring calving cows at two moments of lactation (ML1 and ML2).

	CB-TMR			CB-GRZ			OD-GRZ					*p* Value		
	ML1	ML 2	AT	ML 1	ML 2	AT	ML 1	ML 2	AT	AML 1	A ML2	T	ML	T * ML
FA saturation (g/100 g of fat)														
SFA	65.7 ± 0.5	62.8 ± 0.5	64.3 ± 0.4 ^x^	62.4 ± 0.6	61.4 ± 0.6	61.9 ± 0.4 ^y^	63.4 ± 0.6	61.6 ± 0.6	62.3 ± 0.4 ^y^	63.7 ± 0.3 ^A^	61.9 ± 0.3 ^B^	<0.001	<0.001	ns
UFA	34.2 ± 0.5	37.1 ± 0.5	35.7 ± 0.4 ^y^	37.5 ± 0.6	38.4 ± 0.6	37.9 ± 0.4 ^x^	36.5 ± 0.6	38.2 ± 0.6	37.4 ± 0.4 ^x^	36.1 ± 0.3 ^B^	37.9 ± 0.3 ^A^	<0.001	<0.001	ns
MUFA	29.1 ± 0.5	32.3 ± 0.5	30.7 ± 0.3 ^y^	32.7 ± 0.5	33.7 ± 0.5	33.2 ± 0.4 ^x^	31.8 ± 0.5	33.6 ± 0.5	32.7 ± 0.4 ^x^	31.2 ± 0.3 ^B^	33.2 ± 0.3 ^A^	<0.001	<0.001	ns
PUFA	5.1 ± 0.1	4.8 ± 0.1	4.9 ± 0.1	4.8 ± 0.1	4.7 ± 0.1	4.7 ± 0.1	4.7 ± 0.1	4.6 ± 0.1	4.7 ± 0.1	4.9 ± 0.1	4.7 ± 0.1	ns	0.08	ns
n-3	0.48 ± 0.04 ^bc^	0.45 ± 0.04 ^c^	0.46 ± 0.02 ^y^	0.67 ± 0.04 ^a^	0.61 ± 0.04 ^ab^	0.64 ± 0.03 ^x^	0.43 ± 0.04 ^c^	0.57 ± 0.04 ^abc^	0.51 ± 0.03 ^y^	0.53 ± 0.03	0.54 ± 0.03	<0.001	ns	0.03
n-6	3.4 ± 0.1 ^a^	2.9 ± 0.1 ^b^	3.1 ± 0.07 ^x^	2.6 ± 0.1 ^bc^	2.4 ± 0.1 ^c^	2.5 ± 0.07 ^y^	2.2 ± 0.1	2.3 ± 0.1 ^c^	2.2 ± 0.07 ^y^	2.7 ± 0.05 ^A^	2.5 ± 0.06 ^B^	<0.001	0.03	0.02
n6/n3	7.1 ± 0.3	6.4 ± 0.3	6.8 ± 0.1 ^x^	3.4 ± 0.3	4.7 ± 0.3	3.8 ± 0.2 ^y^	4.4 ± 0.3	3.9 ± 0.3	4.2 ± 0.2 ^y^	4.9 ± 0.2	4.8 ± 0.1	<0.001	ns	0.08
Trans	4.1 ± 0.2	4.2 ± 0.2	4.2 ± 0.1 ^y^	5.1 ± 0.2	5.2 ± 0.2	5.2 ± 0.1 ^x^	5.5 ± 0.2	5.1 ± 0.2	5.3 ± 0.1 ^x^	4.9 ± 0.1	4.8 ± 0.1	<0.001	ns	ns
FA origin (g/100 g of fat)														
De novo														
(C4:0-C15:1)	19.3 ± 0.5	17.2 ± 0.5	18.2 ± 0.3	19.4 ± 0.5	16.1 ± 0.5	17.7 ± 0.3	19.5 ± 0.5	15.9 ± 0.5	17.7 ± 0.3	19.4 ± 0.3 ^A^	16.4 ± 0.3 ^B^	ns	<0.001	ns
Mixed origin														
(C16:0 + C16:1)	37.6 ± 0.5 ^a^	37.4 ± 0.5 ^a^	37.5 ± 0.4 ^x^	33.6 ± 0.5 ^b^	36.6 ± 0.5 ^a^	35.1 ± 0.4 ^y^	33.2 ± 0.5 ^b^	36.5 ± 0.5 ^a^	34.8 ± 0.4 ^y^	34.8 ± 0.3 ^B^	36.8 ± 0.3 ^A^	<0.001	<0.001	0.002
Preformed														
(>C17:0)	43.6 ± 0.7	45.2 ± 0.7	44.2 ± 0.5 ^x^	46.9 ± 0.7	47.1 ± 0.7	47.0 ± 0.5 ^y^	46.5 ± 0.7	47.4 ± 0.7	46.9 ± 0.5 ^y^	45.5 ± 0.7	46.5 ± 0.7	<0.001	0.06	ns
SFA/UFAratio	1.9 ± 0.04	1.7 ± 0.04	1.8 ± 0.03 ^x^	1.6 ± 0.04	1.5 ± 0.04	1.6 ± 0.03 ^y^	1.7 ± 0.04	1.6 ± 0.04	1.6 ± 0.03 ^y^	1.7 ± 0.02 ^A^	1.6 ± 0.02 ^B^	<0.001	<0.001	ns
Atherogenicity index (AI)	2.2 ± 0.05	1.9 ± 0.05	2.1 ± 0.04 ^x^	1.9 ± 0.05	1.8 ± 0.05	1.9 ± 0.04 ^y^	2.1 ± 0.06	1.8 ± 0.06	1.9 ± 0.04 ^y^	2.1 ± 0.03 ^A^	1.9 ± 0.03 ^B^	<0.001	<0.001	ns
Thrombogenicity index (TI)	3.4 ± 0.05	3.0 ± 0.07	3.2 ± 0.05 ^x^	2.9 ± 0.08	2.8 ± 0.07	2.8 ± 0.05 ^y^	3.0 ± 0.07	2.9 ± 0.07	2.9 ± 0.05 ^y^	3.1 ± 0.04 ^A^	2.9 ± 0.04 ^B^	<0.001	0.005	ns

T: treatment; ML: moments of lactation; and T * ML: treatment and moments of lactation interaction effect. Treatments: CB-TMR, CB-GRZ, and OD-GRZ. ML1: 80 ± 15 days in milk (DIM), and ML2: 155 ± 15 DIM. AT: average of both moments of lactation in each treatment; AML1: average of treatments in ML1; and AML2: average of treatments in ML2. Different letters indicate significant difference (*p* < 0.05): ^x,y^ indicate significant difference between AT; ^A,B^ indicate significant difference between AML; and ^a,b,c^ indicate significant difference in treatment by moment of lactation interaction. ns: not significant. SFA: saturated; MUFA: monounsaturated; PUFA: polyunsaturated; and UFA: unsaturated.

**Table 4 animals-13-01426-t004:** Individual fatty acid profilse (FAP) (mean ± SEM) in confined (CB-TMR) and mixed systems with low (CB-GRZ) and high (OD-GRZ) environmental exposure in spring calving cows at two moments of lactation (ML1 and ML2).

	CB-TMR			CB-GRZ			OD-GRZ					*p* Value		
	ML1	ML2	AT	ML1	ML2	AT	ML1	ML2	AT	AML1	AML2	T	ML	T * ML
FA (g/100 g of fat)														
C4:0	0.84 ± 0.03	0.6 ± 0.03	0.72 ± 0.02 ^x^	0.79 ± 0.03	0.52 ± 0.03	0.65 ± 0.02 ^xy^	0.7 ± 0.03	0.52 ± 0.03	0.61 ± 0.02 ^y^	0.77 ± 0.02 ^A^	0.55 ± 0.02 ^B^	0.003	<0.001	ns
C6:0	0.94 ± 0.03	0.73 ± 0.03	0.83 ± 0.02 ^x^	0.85 ± 0.03	0.67 ± 0.03	0.76 ± 0.02 ^y^	0.84 ± 0.03	0.61 ± 0.03	0.72 ± 0.02 ^y^	0.88 ± 0.01 ^A^	0.67 ± 0.01 ^B^	0.001	<0.001	ns
C8:0	0.77 ± 0.02	0.6 ± 0.02	0.68 ± 0.02 ^x^	0.70 ± 0.03	0.52 ± 0.03	0.68 ± 0.02 ^y^	0.66 ± 0.03	0.47 ± 0.03	0.57 ± 0.002 ^y^	0.71 ± 0.02 ^A^	0.53 ± 0.01 ^B^	<0.001	<0.001	ns
C10:0	2.2 ± 0.07	1.7 ± 0.02	1.2 ± 0.05 ^x^	1.9 ± 0.08	1.4 ± 0.08	1.7 ± 0.06 ^y^	1.7 ± 0.08	1.3 ± 0.08	1.5 ± 0.06 ^y^	1.9 ± 0.04 ^A^	1.5 ± 0.04 ^B^	<0.001	<0.001	ns
C12:0	2.5 ± 0.08	2.2 ± 0.08	2.4 ± 0.05 ^x^	2.3 ± 0.08	1.8 ± 0.08	2.1 ± 0.06 ^y^	2 ± 0.08	1.8 ± 0.08	1.9 ± 0.06 ^y^	2.3 ± 0.04 ^A^	1.9 ± 0.04 ^B^	<0.001	<0.001	ns
C14:0	9.7 ± 0.3 ^b^	9.0 ± 0.2 ^bc^	9.3 ± 0.25	9.7 ± 0.3 ^b^	8.5 ± 0.2 ^c^	9.1 ± 0.26	10.8 ± 0.3 ^a^	8.4 ± 0.2 ^c^	9.6 ± 0.3	10.3 ± 0.2 ^A^	8.26 ± 0.2 ^B^	ns	<0.001	0.005
C14:1 cis	0.63 ± 0.05 ^b^	0.81 ± 0.05 ^ab^	0.72 ± 0.03	0.88 ± 0.06 ^a^	0.76 ± 0.06 ^ab^	0.82 ± 0.04	0.64 ± 0.06 ^ab^	0.76 ± 0.06 ^ab^	0.72 ± 0.04	0.72 ± 0.03	0.78 ± 0.03	0.08	ns	0.03
C15:0	1.2 ± 0.04	1.2 ± 0.04	1.2 ± 0.03 ^y^	1.9 ± 0.04	1.5 ± 0.04	1.7 ± 0.03 ^x^	1.7 ± 0.04	1.6 ± 0.04	1.7 ± 0.03 ^x^	1.6 ± 0.02 ^A^	1.46 ± 0.02 ^B^	<0.001	<0.001	ns
C16:0	35.7 ± 0.5 ^a^	35.3 ± 0.5 ^a^	35.5 ± 0.35 ^x^	31.2 ± 0.5 ^b^	34.3 ± 0.5 ^a^	32.7 ± 0.38 ^y^	31.3 ± 0.5 ^b^	34.1 ± 0.5 ^a^	32.7 ± 0.29 ^y^	32.7 ± 0.3 ^B^	34.6 ± 0.3 ^A^	<0.001	<0.001	0.001
C16:1 cis	1.6 ± 0.08 ^ab^	1.8 ± 0.08 ^a^	1.7 ± 0.06	1.8 ± 0.09 ^a^	1.8 ± 0.09 ^a^	1.8 ± 0.06	1.3 ± 0.09 ^a^	1.9 ± 0.09 ^a^	1.6 ± 0.06	1.3 ± 0.05 ^B^	1.9 ± 0.05 ^A^	ns	<0.001	0.001
C16:1 trans	0.33 ± 0.03 ^d^	0.35 ± 0.03 ^cd^	0.35 ± 0.02 ^z^	0.51 ± 0.03 ^b^	0.44 ± 0.03 ^bcd^	0.47 ± 0.02 ^y^	0.75 ± 0.03 ^a^	0.48 ± 0.03 ^bc^	0.62 ± 0.02 ^x^	0.53 ± 0.02 ^A^	0.43 ± 0.02 ^B^	<0.001	<0.001	<0.001
C18:0	10.6 ± 0.43	10.3 ± 0.43	10.5 ± 0.31	11.5 ± 0.45	10.6 ± 0.45	11 ± 0.32	11.5 ± 0.45	11.2 ± 0.45	11.4 ± 0.32	11.2 ± 0.2	10.7 ± 0.02	ns	ns	ns
C18:1 cis	22.8 ± 0.43 ^b^	25.6 ± 0.43 ^a^	24.2 ± 0.31	24.7 ± 0.45 ^a^	25.7 ± 0.45 ^a^	25.2 ± 0.32	24.6 ± 0.45 ^a^	25.7 ± 0.45 ^a^	25.2 ± 0.32	24.1 ± 0.3 ^B^	25.7 ± 0.3 ^A^	0.03	<0.001	0.07
C18:1 trans	3.3 ± 0.15	3.3 ± 0.15	3.3 ± 0.11 ^y^	4.1 ± 0.16	4.4 ± 0.16	4.2 ± 0.11 ^x^	4 ± 0.16	4 ± 0.16	4 ± 0.11 ^x^	3.7 ± 0.09	3.9 ± 0.09	<0.001	ns	ns
C18:2 CLA	0.73 ± 0.06 ^c^	0.82 ± 0.06 ^bc^	0.77 ± 0.04 ^y^	1.0 ± 0.06 ^ab^	1.15 ± 0.06 ^a^	1.1 ± 0.04 ^x^	1.2 ± 0.06 ^a^	1.1 ± 0.06 ^ab^	1.12 ± 0.04 ^x^	0.98 ± 0.03	1.0 ± 0.03	<0.001	ns	0.04
C18:3 (n-3)	0.26 ± 0.03	0.24 ± 0.03	0.25 ± 0.02 ^z^	0.53 ± 0.03	0.46 ± 0.03	0.49 ± 0.02 ^x^	0.39 ± 0.03	0.36 ± 0.03	0.38 ± 0.02 ^y^	0.39 ± 0.02	0.35 ± 0.02	<0.001	ns	ns

T: treatment; ML: moment of lactation; and T * ML: treatments and moment of lactation interaction. Treatments: CB-TMR, CB-GRZ, and OD-GRZ. ML1: 80 ± 15 days in milk (DIM), and ML2: 155 ± 15 days in milk (DIM). AT: average of both moments of lactation in each treatment; AML1: average of treatments in ML1; and AML2: average of treatments in ML2. Different letters indicate significant difference (*p* < 0.05): ^x,y,z^ indicate significant difference between AT; ^A,B^ indicate significant difference between AML; and ^a,b,c,d^ indicate significant difference in treatment and moment of lactation interaction. ns: no significance.

**Table 5 animals-13-01426-t005:** Fatty acid profiles (FAP) of milk and cheese (mean ± SEM) in confined (CB-TMR) and mixed systems with low (CB-GRZ) and high (OD-GRZ) environmental exposure in spring calving cows.

	MilkP				Cheese			
	CB-TMR	CB-GRZ	OD-GRZ	*p* Value	CB-TMR	CB-GRZ	OD-GRZ	*p* Value
				T				T
FA saturation (g/100 g of fat)								
SFA	63.5 ± 0.9	64.5 ± 0.9	64.1 ± 0.9	ns	64.2 ± 0.5	64.4 ± 0.5	64.7 ± 0.5	ns
MUFA	32.3 ± 0.7	31.2 ± 0.7	31.6 ± 0.7	ns	31.8 ± 0.5	31.5 ± 0.5	31.3 ± 0.5	ns
PUFA	4.0 ± 0.2	3.0 ± 0.2	4.1 ± 0.2	ns	3.8 ± 0.1	3.8 ± 0.1	3.8 ± 0.1	ns
n-3	0.26 ± 0.03 ^x^	0.51 ± 0.03 ^y^	0.50 ± 0.03 ^y^	<0.001	0.43 ± 0.03 ^x^	0.55 ± 0.03 ^y^	0.46 ± 0.03 ^xy^	<0.001
n-6	2.5 ± 0.1 ^x^	1.9 ± 0.1 ^y^	1.85 ± 0.1 ^y^	<0.001	2.9 ± 0.1 ^x^	2.6 ± 0.1 ^y^	2.3 ± 0.1 ^y^	<0.001
n6/n3	9.5 ± 0.1 ^x^	3.7 ± 0.1 ^y^	3.7 ± 0.1 ^y^	<0.001	7.1 ± 0.5 ^x^	4.8 ± 0.5 ^y^	5.2 ± 0.5 ^y^	<0.001
Trans	3.5 ± 0.1	4.2 ± 0.1	2.0 ± 0.1	<0.001	3.6 ± 0.1	3.8 ± 0.1	3.9 ± 0.2	ns

MilkP: milk from pen. T: treatment. Treatments: (CB-TMR, CB-GRZ, and OD-GRZ). Different letters indicate significant difference (*p* < 0.05). ^x,y^ indicate significant difference between treatment (CB-TMR, CB-GRZ, and OD-GRZ). ns: no significance. SFA: saturated; MUFA: monounsaturated; and PUFA: polyunsaturated.

## Data Availability

Data Availability Statement: Data are available upon reasonable request to the corresponding author.

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
