# Peer review of "Different Conditions during Confinement in Pasture-Based Systems and Feeding Systems Affect the Fatty Acid Profile in the Milk and Cheese of Holstein Dairy Cows"

_animals, 2023, doi:10.3390/ani13081426_

Round 1

Reviewer 1 Report

The manuscript is good, with the results clear, indicating that feeding is the most important factor changing milk composition, namely the inclusion of pasture. 

Although the results were as expected, the milk and rennet analyses (in a second phase) once again proved the importance of grazing in milk production. 

A detailed analysis of the feed provided to the animals is still lacking. We know that pastures have different characteristics throughout the season. The work lacks analysis to determine the nutritional value of food. For me, this is the great weakness of this work.

Author Response

Review 1 (R1): Manuscript review
Manuscript ID:
animals-2321802

Title: Different Conditions During Confinement (Compost-Bedded 2 Pack Barn vs. Outdoor Soil- Bedded Pen) in Mixed Systems 3 (Grazing Plus Total Mixed Ration) And Feeding Systems 4 (Mixed vs. Confined) Affect Fatty Acid Profile in Milk and 5 Cheese of Holstein Dairy Cows

General Observations:

The manuscript is good, with the results clear, indicating that feeding is the most important factor changing milk composition, namely the inclusion of pasture.

Authors (A): We thank the reviewer for the positive comments on the manuscript.

R1: Although the results were as expected, the milk and rennet analyses (in a second phase) once again proved the importance of grazing in milk production.

A detailed analysis of the feed provided to the animals is still lacking. We know that pastures have different characteristics throughout the season. The work lacks analysis to determine the nutritional value of food. For me this is the great weakness of this work.

A: We thank the reviewer for the time spent reviewing the manuscript and for the suggestions, which are answered in the specific comments below.

R1: Title:  Title is too long and should be shortened. What is between parentheses can be removed or the text can be modified to put this information directly in the title.

A: We thank the reviewer for the suggestion to shorten the title. We change the title as follows:

“Different condition during confinement in pasture-based systems and feeding systems affect fatty acid profile in milk and cheese of Holstein dairy cows”

R1: Simple Summary: In the Simple Summary, in accordance with the instructions for authors, "No references are cited and no abbreviations.” Therefore, you must rewrite this part of the work, to eliminate the abbreviations. I also recommend that you write in such a way that a layperson can understand the work and that you clearly indicate the objectives and conclusions of the work.

A: We thank for the comments of Simple Summary. We rewrote the Simple Summary according to you suggestion:

Simple Summary: “Pasture-based systems have advantages compared to confined systems such as higher proportion of beneficial milk fatty acids for consumers. However, in mixed systems (grazing+total mixed ration) the cows are more exposed to external climatic conditions. Due to climate change (high temperature or heavy rains), cows spend more time in confinement facilities where supplementation is offered, mainly in intensive mixed systems (high stocking rate). Therefore, the facilities conditions during confinement acquire great importance. To our knowledge, no studies have evaluated how the different facilities conditions during confinement in mixed systems affect the fatty acid profile in milk and cheese. The objective of this study was to compare the fatty acid profile between mixed systems with different conditions during confinement [compost-bedded pack barn vs. outdoor soil-bedded pen] and between mixed systems and confinement system (100% total mixed ration)] in compost-bedded pack barn]. In conclusion, the compost-bedded pack barn during confinement in mixed systems ensued better milk quality (higher percentage of n-3 and C18:3 in milk) compared to outdoor soil-bedded pen. However, the fatty acid profile in milk, pooled milk (MilkP) and cheese were affected to a greater extent by feeding management than by controlling the environment during confinement.”

R1: Summary:

According to the instructions for authors, the abstract must contain the following structure: 1) Background: Place the question addressed in a broad context and highlight the purpose of the study; 2) Methods; 3) Results; and 4) Conclusion.

In this case, the summary does not have a background and goes directly to the objectives. 1

You must write a line with a background. Note that you are at the maximum word limit (200 words) so as not to exceed them.

A: We added a phrase of background and modify the text to not exceed 200 words

We added: “The diet of dairy cows influences the fatty acid (FA) profile of milk and cheese, but how they are affected by different conditions during confinement in mixed systems (MS:grazing+total mixed ration:TMR) is not known.” Line: 34-36.

The new text is: “The diet of dairy cows influences the fatty acid (FA) profile of milk and cheese, but how they are affected by different conditions during confinement in mixed systems (MS:grazing+total mixed ration:TMR) is not known. The aim of this study was to compare the FA of milk and cheese from MS in a compost-bedded pack barn (CB-GRZ) versus an outdoor soil-bedded pen (OD-GRZ) during confinement, and with a confinement system (100%TMR) in a compost-bedded pack barn (CB-TMR). Individual milk samples (n=12 cows/group), cheese and pooled milk (MilkP) samples were collected. The saturated FAs percentages in milk and the n-6/n-3 ratio in MilkP and cheese were greater in CB-TMR (p<0.0001), while unsaturated and monounsaturated FAs percentages in milk were lower in CB-TMR than in MS (p<0.001). Milk n-3, C18:3, and conjugated linoleic acid percentages were lower in CB-TMR than in MS (p<0.001). Milk n-3 and C18:3 were higher in CB-GRZ than in OD-GRZ (p<0.01), but no differences were observed between MS in MilkP and cheese. In conclusion, CB-GRZ cows during confinement produced better quality milk compared to OD-GRZ. However, the FA profiles of milk, MilkP, and cheese were affected to a greater extent by feeding management than by the conditions during confinement.”

R1: Keywords:

Two of the keywords are repeated in the title. It is recommended that you replace them with other words.

A: Thanks for this comment. Based on the previous suggestion of the reviewer, we changed some words in Title so, now key words are not repeated in the title.  

In the Title we removed: “(Compost-Bedded Pack Barn vs. Outdoor Soil- Bedded Pen)”; “Mixed Systems”; “(Grazing Plus Total Mixed Ration)”; “(Mixed vs. Confined)” and “in pasture-based systems” was added.

Title final is: “Different condition during confinement in pasture-based systems and feeding systems affect fatty acid profile in milk and cheese of Holstein dairy cows”

Key words are: dairy products; lipid; bovine; grazing; total mixed ration; climatic condition 

R1Introduction:

Between line 69 and line 74: animals in extreme weather conditions are said to suffer from stress. However, they should make a counterpoint, saying that animals in pasture systems, in temperate climates, can be on pasture all year round, as is the case in the Azores, Portugal or New Zealand, with milk derivatives having higher levels of carotenes, for example.

Therefore, I recommend that you make a counterpoint about these pasture systems and that you mention that only in areas where the climates are extreme, animals should be sheltered during the cold or extremely hot seasons.

A: We appreciate your comments, but we have some considerations about it. The climatic change has given extreme phenomena in temperate countries, due to extreme temperatures in summer, frost in spring, drought or floods at times where they did not occur before. This is mainly a problem in countries where the cows are in pasture (grazing) year-round, a problem explained in detail by Hennessy et al. (2022), for European temperate climate countries.

However, taking into account the reviewer's suggestion we made some changes and added the following information:

“In countries with temperate climates, cows are grazing throughout the year, resulting in a better nutritional quality of milk than confined systems (i.e., O´Callaghan et al., 2016; Alothman et al., 2019). However, climate change has caused more extreme weather events, such as increased temperatures in summer, droughts, frosts in spring, heavy rains, which affects the pastures growth and time available for grazing. Therefore, these climatic phenomena become more important in temperate countries (Hennessy et al., 2020 [21]; Fariña and Chilibroste, 2019 [22], Méndez et al., 2023 [23]).” Lines: 75-80

R1: In addition, he mentions physiological changes when animals are exposed to extreme situations. What are these changes?

A: We added: “(energy demand for thermoregulation, decreased dry matter intake, decreased milk production)” and we added some references: Van Laer et al., 2015 [25]; Kadzere et al., (2002) [26]; Moretti et al., 2017 [27]. Lines 81-82.

R1: Material and methods

Item 2.2.1 line 173

The times that refer to milking is always “am”, one of them should be pm (I assume it is 3p.m). You should correct it.

A: We thank the reviewer for pointing out this writing error. We changed “am” to “pm” Line 190. We realized a mistake in milking times. We changed “5:00 to 4:00” and “3:00 to 5:00”.

Therefore, the information about milking in the manuscript is: “All cows were milked twice a day (4:00 am and 5:00 pm) and individual milk yield was individually recorded.” Line 190

R1: To enrich the article, it would be interesting to put the analyses done to the feed (pasture and TMR). If you have this data, please publish it (it could be as a supplementary table).

A: Table 1 shows information about analysis of pasture and TMR in the three moments which the samples were taken (CP: crude protein; NDF: neutral detergent fiber; ADF: acid detergent fiber). Between lines 168 and 165 information about the pasture used in different moments of experiment is presented.

However, considering the reviewer's suggestion we added information on Herbage allowance (HA) (kgDM/cows/day) in every moment (Section 1: ML1 and ML2) and Section 2.

We added: “Herbage allowance (HA) were of 28 kg DM/cow/day, 46 kg DM/cow/day and 31 kg DM/cow/day; respectively”.  Lines 170-171.

R1. In the statistical analysis, it remains to indicate which tests were used to verify the assumptions for the use of ANOVA. Most of the data are percentages, which implies that they make a data transformation. What is the transformation that was done?

Should all this information be in the material and methods?

A: We appreciate the suggested for the reviewer and we added information about tests which were used to verify the assumptions for the use of ANOVA.

In Statistical Analysis (2.4) (manuscript) we added the following information: “The Shapiro test was used to evaluate the normal distribution of the data. The goodness of fit of each model was checked by visual inspection of the residuals. The election of final models considered the Akaike’s information criterion AIC.” Lines 281-283.

The data expressed as percentages (%) are not transformed, they only express the grams of each fatty acid per 100 grams of total fat, as has been used and expressed in several previous reports:

  • Vibart et al., 2017: Milk production and composition, nitrogen utilization, and grazing behavior of late-lactation dairy cows as affected by time of allocation of a fresh strip of pasture. Journal of Dairy Science,100 (7), 5305-5318. https://doi.org/10.3168/jds.2016-12413
  • Barca, J. et al., (2018). Milk fatty acid profile from cows fed with mixed rations and different access time to pastureland during early lactation. Journal of Animal Physiology and Animal Nutrition, 102(3), 620–629. https://doi.org/10.1111/jpn.12826
  • O´Callagham et al., 2019: “Influence of Supplemental Feed Choice for Pasture-Based Cows on the Fatty Acid and Volatile Profile of Milk” Foods, 8, 137. doi:10.3390/foods8040137
  • Pastorini et al., 2019: Productive performance and digestive response of dairy cows fed different diets combining a total mixed ration and fresh forage. Journal of Dairy Science, 102(5), 4118–4130. https://doi.org/10.3168/jds.2018-15389
  • Grille et al., 2022: “Milk Fatty Acid Profile of Holstein Cows When Changed from a Mixed System to a Confinement System or Mixed System with Overnight Grazing,” Int J Food Sci, doi: 10.1155/2022/5610079.

This information is in the original version of manuscript. “Milk fat composition is expressed as grams of each individual FA per 100 g of total FA” Lines: 245-246.

R1: Results

The formulas presented in the legends should be included in the materials and methods. Figure legends should contain only the essential data.

A: We appreciate this comment, but we think the Tables must be “Self Explanatory”, so all information must be explained in the table and figure footer. For this reason, we would prefer to keep that information in the legends if the editor and the reviewer consider it correct.

However, we consider the reviewer's suggestion and we did the following changes:

In legend of Table 3, were removed the formulas (Atherogenicity index (AI) calculated as (12:0 + 4 x 14:0 + 16:0/ (MUFA + PUFA); Thrombogenicity index (TI): (C14:0+C16:0+C18:0) (0.5×MUFA+0.5×(n−6) +3 ×(n−3)/(n−6) and they were included in materials and methods. Lines 248-250.

In legend of Figure 2: we removed “[ML1: 80±15 days in milk (DIM), ML2: 155±15 DIM]”. Line: 230. This information is in material and methods. Line: 192-193

R1: Discussion

I understand the use of acronyms to abbreviate some words (as for example TMR) however the discussion is difficult to read, because there is an excess of acronyms used and we have to verify what each one of them means. They are treatments, lactation times, systems, etc...

I suggest you simplify a little the language in the discussion, using only internationally known acronyms.

A: We changed “MS” to “mixed systems”. Lines: 436, 438, 441, 445, 450, 463, 471, 584, 535.

We changed “FAP” to “fatty acid profile”. Lines: 444, 449, 453, 467, 470, 473, 509, 529, 534, 541, 542.

R1: Conclusion

Simple and straightforward. It's good.

A: We thank the reviewer for the positive comments and for highlighting the strengths of the conclusion

Reviewer 2 Report

Dear authors, 

some suggestions are reported (attached word)

Thanks 

Best regards

Author Response

Reviewer 2 (R2). Simple summary

Line 26. TMR is the acronym, ok, a a full description is required before the acronym.

Authors (A): the acronym was removed and a full description “total mixed ration” was written in Line 21.

R2: Line 20. The square bracket is open only.

A: We appreciate the observation of the reviewer. The change was done (line 29)

R2: Introduction

Lines 70-71. Are rain, wind and solar radiation extreme weather conditions? Are you sure??

A: We thanks to reviewer for this comment with respect to weather condition.

Based on some comments from R1, we made changes at the beginning of the paragraph where was added more information about extreme weather conditions.

Between line 76 - 80 we added “However, climate change has caused more extreme weather events, such as increased temperatures in summer, droughts, frosts in spring, heavy rains, which affects the pastures growth and time available for grazing. Therefore, these climatic phenomena become more important in temperate countries.”

Line 80: “such as rain, wind and solar radiation” was removed

R2: Line 72. After the sentence: Cows can suffer from heat stress in spring-summer, which impacts milk production and quality I suggest bibliography: Estimation of maximum thermos-hygrometric index thresholds affecting milk production in Italian Brown Swiss cattle. Maggiolino et al. (2020), 103 (9), 8541-8553 https://doi.org/10.3168/jds.2020-18622.

Line 74. I suggest to introduce the theme of animal welfare, helpful to increase the milk yield. I propose as literature source, a research reflection: Napolitano F, Bragaglio A, Sabia E, Serrapica F, Braghieri A and De Rosa G. The human−animal relationship in dairy animals. Journal of Dairy Research (2020). https:// doi.org/10.1017/S0022029920000606.

A: We appreciate the suggest of reviewer to better the introduction with other articles about heat stress and welfare. We added both articles. First reference was added line 85 [32]. Beside we added “on welfare” before second reference [28]. Line: 84.

R2: Lines 87-88. You adopted the word roughage, I suggest a better word. Thanks.

A: We appreciate the suggestion of the reviewer, so we changed “roughage” by “conserved forage” Line: 99.

R2: Lines 90-91. As still indicated. Do you consider extreme weather conditions the rain?

Please explain it.

A: We thank the comments of reviewer. We consider that the heavy rain is an important problem in countries where pasture- based system are predominant. In countries with temperate climates where cows are grazing during all year (Mixed systems: grazing + TMR), heavy rains and high temperatures in summer (ITH above 72, Cruz y Saravia, 2008; Román et al., 2019) can affect the animal welfare and decrease time available for grazing, so the cows spend more time in confinement. In addition, heavy rains can affect the facilities (places where cows are when aren´t grazing), e.i: a lot of mud which impairing welfare (decreased lying time; Chen et al., 2017; Fisher et al., 2003; Shütz et al., 2019)

In the manuscript we added:

Line 102: “heavy rains”

Line 103: “which could affect welfare” (Chen et al., 2017 [39]; Roman et al., 2019 [40]).

R2: Lines 91-93. In Uruguayan dairy systems...areas of cows. This phrase is unclear, improve it please. Thanks.

A: We thank the reviewer for the comment.

We removed: “Uruguayan dairy systems, although 75% of the farms have natural shade, it is usually located on the roads and not in resting and feeding areas of cows.”

We changed this sentence: “In most dairy systems, the cows are confined in open sky paddocks when they are not grazing [31]” by this: “Most dairy systems in Uruguay keep cows confined in open sky paddocks when they are not grazing, without shade on resting and feeding areas [41]." Lines 103, 109, 110

R2: Materials and methods

Line 125. Maybe breed indications (Holstein) should be added.

A: We added “Holstein”. Line 140.

R2: Line 145. What kind of hay? Species, chemical characteristics...

A: We added: “Fescue” in line: 160

R2: Line 147. Is the fat percentage a mean value or the percentage was obtained with a equation (e.g. similarly to fat and protein corrected milk formula, adopted to normalize the milk)?

A: The data expressed as percentages (%) they only express the grams of each fatty acid per 100 grams of total fat (are not transformed=, as has been used and expressed in several previous reports:

  • Vibart et al., 2017: Milk production and composition, nitrogen utilization, and grazing behavior of late-lactation dairy cows as affected by time of allocation of a fresh strip of pasture. Journal of Dairy Science,100 (7), 5305-5318. https://doi.org/10.3168/jds.2016-12413
  • Barca, J. et al., (2018). Milk fatty acid profile from cows fed with mixed rations and different access time to pastureland during early lactation. Journal of Animal Physiology and Animal Nutrition, 102(3), 620–629. https://doi.org/10.1111/jpn.12826
  • O´Callagham et al., 2019: “Influence of Supplemental Feed Choice for Pasture-Based Cows on the Fatty Acid and Volatile Profile of Milk” Foods, 8, 137. doi:10.3390/foods8040137
  • Pastorini et al., 2019: Productive performance and digestive response of dairy cows fed different diets combining a total mixed ration and fresh forage. Journal of Dairy Science, 102(5), 4118–4130. https://doi.org/10.3168/jds.2018-15389
  • Grille et al., 2022: “Milk Fatty Acid Profile of Holstein Cows When Changed from a Mixed System to a Confinement System or Mixed System with Overnight Grazing,” Int J Food Sci, doi: 10.1155/2022/5610079.

This information is in the original version of manuscript. “Milk fat composition is expressed as grams of each individual FA per 100 g of total FA” Lines: 245-246

R2: Line 173. All cows were milked twice a day (5:00 am and 3:00 am). Are you sure? Maybe am and pm?

A: We thank the reviewer for pointing out this writing error. We changed “am” to “pm” line 189. We realized a mistake in milking times. In line 190 we changed “5:00 to 4:00” and “3:00 to 5:00”.

Therefore, the information about milking in the manuscript is: “All cows were milked twice a day (4:00 am and 5:00 pm) and individual milk yield was individually recorded.” Line: 190

R2: Line 214. The determination of milk fat yield was using. The grammar must be improved. The sentence, written in this way, is misleading.

A: We appreciate this comment of reviewer. We changed the grammar. In Line 233 “the determination of” was removed and it was added: “Milk fat content (%) was determined”

R2: Lines 235 and 236. AOAC. The method must be described, moreover the bibliography must be improved. It is scarce.

A: We thanks to the reviewer for this comment.

In Line 256 we added: “digestion with sulfuric acid and subsequent distillation and titration”

In Line 257-258 we added: “used α-amylase and both (NDF y ADF)”

In Line 259 we added: “according to Méndez et al., (2023) [23]”

R2: Discussion

Line 374-375. Redundant.

A: We removed the sentence: “Furthermore, differences in fat and milk yield were observed only between CB-TMR and mixed systems, but not between the latter (CB-GRZ and OD-GRZ)”. Lines: 439-441.

R2: Lines 397-398. The phrase between brackets is unclear.

A: We removed: “(recommended to be below 4/1)” and now we add the following: “(the recommended value should be below 4/1;)” Lines: 458-459.

R2: Line 433. I propose a different presentation: ...could be due to (i) no high THI... (ii) OD-GRZ facilities...

A: We appreciate the reviewer´s comment, and we did changes according to suggestions:

In line 515, we removed “Firstly” and now we added: (i). In line 517 we removed Secondly, and we added (ii)

R2: Line 438. You can delete “in this sense”.

A: The changed was done in line 520. 

Round 2

Reviewer 2 Report

Dear authors

I indicate some suggestions. In accordance with my revisions you improved the manuscript. However I still found something to optimise.

Line 154. Please, give a better and more detailed description of fescue hay

Line 250. NDF y ADF is Spanish. Please put NDF and ADF

You did not answer the question: Have you adopted an equation to estimate the fat milk? similarly to fat and protein corrected milk (FPCM) used to normalize the milk. 

Thanks

Author Response

Dear Ewelina Bik

Assistant Editor

We greatly appreciate the reviewer´s comments and suggestions. We present a new version of the manuscript ID: 2321802

We hope that you find our responses satisfactory and that the manuscript is now acceptable for publication.

Following are our detailed answers:

Reviewer (R): I indicate some suggestions. In accordance with my revisions you improved the manuscript. However, I still found something to optimise.

Author (A): We are very grateful for the comments which helped to improve the manuscript.

R: Line 154. Please, give a better and more detailed description of fescue hay

A: We added the following information for all the ingredients used in the TMR. Lines: 149-152.

“The total mixed ration was composed of whole plant sorghum or corn silage (CP: 5.9, 5.8%; NDF: 51.5, 41.7%; ADF: 30.7, 22.5%; respectively), rye grass silage (CP: 10.9%; NDF: 49.6%; ADF: 29.9%), fescue hay (CP: 9.7%; NDF: 68.3%; ADF: 37.4%), and a commercial concentrate mix (CP: 25.5%; NDF: 34.5%; ADF: 13.5%)”.

R: Line 250. NDF y ADF is Spanish. Please put NDF and ADF

A: Abbreviations were already wright in the correct form in the previous version (please see lines 244-245-247-248)

R: You did not answer the question: Have you adopted an equation to estimate the fat milk? similarly to fat and protein corrected milk (FPCM) used to normalize the milk. 

A: We have answered that question. Our previous response was as follows: “the data expressed as percentages (%) they only express the grams of each fatty acid per 100 grams of total fat (are not transformed) as has been used and expressed in several previous reports.”

Apologies, if perhaps we did not understand the question correctly, because it caused us doubts what you meant by it, or our answer wasn’t clear enough.

Regarding your question "Have you adopted an equation to estimate the fat milk? similarly to fat and protein corrected milk (FPCM) used to normalize the milk." Our answer is no, we have not used an equation to estimate milk fat.

However, we included the information about Fat-corrected milk (FCM 3.5%) in material and methods section, Lines: 183-184:

Fat-corrected milk (FCM) yield of standardized at 3.5% fat, was calculated according to Pastorini et al. [48] and Mendoza et al. [49]

This information was included in Table 2, under the Milk yield row (see following page)

R: Thanks

A: Thank you very much for your comments
